# Persistent Mesodermal Differentiation Capability of Bone Marrow MSCs Isolated from Aging Patients with Low-Energy Traumatic Hip Fracture and Osteoporosis: A Clinical Evidence

**DOI:** 10.3390/ijms25105273

**Published:** 2024-05-12

**Authors:** Mei-Chih Wang, Wei-Lin Yu, Yun-Chiao Ding, Jun-Jae Huang, Chin-Yu Lin, Wo-Jan Tseng

**Affiliations:** 1Biomedical Technology & Device Research Laboratories, Industrial Technology Research Institute, Hsinchu 31057, Taiwan; mcwang@itri.org.tw (M.-C.W.); yuweilin@itri.org.tw (W.-L.Y.); s46024014@itri.org.tw (Y.-C.D.); junjaehuang@itri.org.tw (J.-J.H.); 2Department of Biotechnology and Pharmaceutical Technology, Yuanpei University of Medical Technology, Hsinchu 300102, Taiwan; 3Department of Biomedical Sciences and Engineering, Tzu Chi University, Hualien 97004, Taiwan; 4Institute of New Drug Development, College of Medicine, China Medical University, Taichung 40402, Taiwan; 5Department of Orthopedic Surgery, National Taiwan University Hospital Hsin-Chu Branch, Hsinchu 300195, Taiwan; 6Department of Biological Science and Technology, National Yang Ming Chiao Tung University, Hsinchu 300093, Taiwan

**Keywords:** low-energy trauma, hip bone fracture, osteoporosis, bone marrow-derived stem cell (BMSCs), osteogenesis

## Abstract

A low-energy hit, such as a slight fall from a bed, results in a bone fracture, especially in the hip, which is a life-threatening risk for the older adult and a heavy burden for the social economy. Patients with low-energy traumatic bone fractures usually suffer a higher level of bony catabolism accompanied by osteoporosis. Bone marrow-derived stem cells (BMSCs) are critical in osteogenesis, leading to metabolic homeostasis in the healthy bony microenvironment. However, whether the BMSCs derived from the patients who suffered osteoporosis and low-energy traumatic hip fractures preserve a sustained mesodermal differentiation capability, especially in osteogenesis, is yet to be explored in a clinical setting. Therefore, we aimed to collect BMSCs from clinical hip fracture patients with osteoporosis, followed by osteogenic differentiation comparison with BMSCs from healthy young donors. The CD markers identification, cytokines examination, and adipogenic differentiation were also evaluated. The data reveal that BMSCs collected from elderly osteoporotic patients secreted approximately 122.8 pg/mL interleukin 6 (IL-6) and 180.6 pg/mL vascular endothelial growth factor (VEGF), but no PDGF-BB, IL-1b, TGF-b1, IGF-1, or TNF-α secretion. The CD markers and osteogenic and adipogenic differentiation capability in BMSCs from these elderly osteoporotic patients and healthy young donors are equivalent and compliant with the standards defined by the International Society of Cell Therapy (ISCT). Collectively, our data suggest that the elderly osteoporotic patients-derived BMSCs hold equivalent differentiation and proliferation capability and intact surface markers identical to BMSCs collected from healthy youth and are available for clinical cell therapy.

## 1. Introduction

The primary function of bones is to support the human body, protect the internal organs, and possess the capability of self-healing. Bone mass reaches its maximum at approximately 35 years old, with females reaching their peaks earlier than males, gradually diminishing with age [1,2]. Principally, faster bone mass loss in postmenopausal women increases the risk of osteoporosis and subsequent osteoporotic bone fractures [3]. Notably, the hip joint is a vital weight-bearing joint responsible for the weight transmission of the body trunk to the lower limbs. Once the hip joint fractures, over 90% of patients need to receive surgical treatment, including complex fixations or joint replacement, to recover the maximum ≈80% of original motion capability, even in optimal circumstances with advanced operation and sophisticated devices. In one extreme circumstance, the patients suffered osteoporosis and emerged as a repeated hip fracture; the recovery rate decreased to below 50%, seriously influencing the daily motion and living capability, resulting in heavy social care and family economic burden [4].

According to national insurance data, the rate of hip fractures in Taiwan is first and ninth in Asia and the world ranking, respectively, reaching ≈20,000 patients per year, and most cases result from osteoporosis [5]. Moreover, hip fracture is a life-threatening disease; 20% of patients died one year after the hip fracture, and 80% of patients’ disabilities emerged as bedridden, accompanied by urinary and pulmonary chronic infections [6]. Therefore, an osteoporotic fracture is a critical problem for personal health and the social economy, and government administrators need to address it seriously. Clinical medications for osteoporosis therapy, including bisphosphonate osteoclast inhibitors, monoclonal antibodies targeting the receptor activator of nuclear factor (RANK), selective estrogen-receptor modulators, and recombinant parathyroid hormone osteoblast stimulators, were used [7,8]. However, tissue concentrate or cell transplantation has emerged as a popular alternative treatment for aging-related degenerative diseases, such as platelet-rich plasma (PRP) infusion or stem cell transplantation for osteoarthritis and disc degenerative disease therapy [9,10,11,12], which needs Taiwan Food and Drug Administration (TFDA) further approval.

Bones possess the self-renewing capability to maintain homeostasis between bony anabolism and catabolism, creating a healthy bony microenvironment. Furthermore, bone marrow-derived stem cells play a crucial role in bone regeneration and bone mass homeostasis, creating a healthy bone structure. Mesenchymal stem cells (MSCs) exist in various connective tissues, including fat and bone marrows, and are capable of differentiating into mesodermal cell lineages, including adipocyte, chondrocyte, and osteoblast [13], the paraxial mesodermal-derived tenocyte [14], and multi-typed cells in the dermis, bone, and skeletal muscle [15]. Thanks to their multipotent differentiation capability, MSCs have been recognized as the most promising cell type for tissue engineering and have been demonstrated to be proliferated in vitro and transplanted into damaged animal models in vivo for tissue regeneration [16].

Furthermore, MSCs possess the functions of immunomodulation and immune privilege, which makes them hold the superior function of promoting tissue regeneration compared to mature somatic cells [17]. The mesenchymal stem cells isolated from bone marrow have been demonstrated to differentiate into osteoblasts and be involved in bone remodeling. However, many publications demonstrated that the size of the MSC pool and colony-forming unit (CFU) of MSCs changed and gradually decreased with age [18,19]. Meanwhile, the proliferative capability of MSCs derived from the elders also decreases [20]. Furthermore, the MSCs isolated from the elders tend to differentiate into adipocyte rather than bone reconstructive cell lineages [19,21,22]. Furthermore, the BMSCs and osteoblasts reveal the scenario of senescence [23] and the modified bone tissue microenvironment in the elders [24].

In order to clarify whether the BMSCs isolated from aging and osteoporotic patients remain to possess the innate capability of a healthy stem cell, we collected the BMSCs from hip fracture patients with low-energy-elicited trauma and analyzed the capability of multipotent differentiation and proliferation compared with the BMSCs from youths. These aging patients who incurred hip fractures resulting from low-energy trauma, such as falling while walking, generally also suffered osteoporosis. Accordingly, the BMSCs derived from the osteoporotic patient express inferior osteogenic differentiation tendency; whether this characteristic alters the capability of bone matrix formation and new bone regeneration and further worsens the osteoporosis is not yet to be proved.

Low-energy trauma-elicited bone fractures represent a sign of bone fragility and ongoing osteoporosis, which quickly emerged as a life-threatening hip fracture, resulting in a heavy social burden. BMSCs are critical in osteogenesis, leading to metabolic homeostasis in the healthy bony microenvironment. Notably, patients with low-energy traumatic bone fractures usually suffer a higher level of bony catabolism accompanied by osteoporosis. However, whether the BMSCs derived from the patients who suffered osteoporosis and low-energy traumatic hip fractures preserve a sustained mesodermal differentiation capability, especially in osteogenesis, is yet to be explored in a clinical setting. Therefore, we collected BMSCs from clinical hip fracture patients accompanied by evidenced osteoporosis in the current study. The CD marker identification, cytokine examination, and osteogenic and adipogenic differentiation were evaluated and compared with the BMSCs withdrawn from the healthy young donors.

## 2. Results

### 2.1. Details of OP Patient-Derived hBMSCs

Patients recruited in the current study were older than 50 years with low-energy hip fractures, and all patients’ Informed Consent Forms (ICFs) were collected before the study. The National Taiwan University-affiliated hospital, Hsin-Chu, conducted the clinical study from May 2019 to September 2020 under IRB protocol approval numbers: 108-006-E and 103-018-F. We randomly recruited 19 patients, and the details of patients’ disease backgrounds revealed that many patients suffered from varied chronic diseases and were under composite medications (Appendix A). After bone marrows were withdrawn from patients, which were immediately subjected to serial Ficoll sedimentation, isolation, and purification (Appendix A), the P2 hBMSCs were cryopreserved according to the standard protocol, and the P3–P7 hBMSCs were used for the subsequent examinations. In the 19 samples (Appendix A), OP-005, OP-007, OP-008, OP-011, and OP-015 were excluded due to patients’ HBV or HCV infection; OP-012 and OP-013 were excluded due to hBMSCs’ insufficient proliferation and bacteria contamination, respectively; and the patient’s midterm secession terminated OP-006. We collected hBMSCs from 11 elderly osteoporotic patients (Appendix A) and hBMSC specimens from 3 healthy younger donors (Appendix A) for subsequent experimental comparison.

### 2.2. OP Patient-Derived hBMSCs Reserved the Osteogenic and Adipogenic Differentiation Capability

Mesodermal-oriented differentiation, including adipogenesis and osteogenesis, is the vital commitment of hBMSCs to participating in tissue healing, such as bone regeneration. Previous publications revealed that the BMSCs derived from elders or osteoporotic patients tend to differentiate into adipocytes rather than osteocytes [21,25]. The changed intrinsic differentiation capability and microenvironment of bone marrow cavities resulted in the disturbed homeostasis between adipogenesis and osteogenesis of BMSCs, leading to insufficient osteogenic orientation and subsequent osteoporosis. In order to evaluate the osteogenic and adipogenic tendency of hBMSCs collected from elderly osteoporotic patients, the hBMSCs were cultured in osteogenic and adipogenic induction medium for differentiation capability examinations, respectively. Our data reveal that the osteoporotic hBMSCs exhibit an apparent osteogenic differentiation (Figure 1A) with a significant calcium deposition capability (*p* < 0.001) (Figure 1B); meanwhile, the adipogenic differentiation was maintained (Figure 1C), showing a significant oil droplet formation compared with hBMSCs without induction medium (*p* < 0.05) (Figure 1D). These data are distinct from the previous publications, which depict an inferior osteogenic tendency. However, our data demonstrate the reserved adipogenic differentiation capability without compromising the osteogenic differentiation tendency.

### 2.3. Surface Marker and Cytokine Secretion Analysis of OP Patient-Derived hBMSCs

To identify the surface markers of hBMSCs that fit the standards defined by the International Society of Cell Therapy (ISCT) [26], the markers of hBMSCs collected from elderly osteoporotic patients and healthy younger donors were analyzed by flow cytometry. The positive markers, such as CD73, CD90, and CD105, and the negative markers, such as CD14, CD19, CD34, CD45, and HLA-DR, were analyzed, with values > 95% and ≤2% for the positive and negative markers, respectively, considered qualified. The representative flow cytometric diagram shows the apparent peak shifts of CD73, CD90, CD105, and HLA-ABC (Figure 2A). The means of CD73 and CD90 are 95.7% and 98.2%, respectively. Almost all BMSCs express moderate to high levels of human leukocyte antigen (HLA) class I molecules (HLA-ABC), which demonstrate multipotent differentiation capability [27]. Nevertheless, the CD105 is not qualified, possibly due to an osteoporotic hBMSC’s value being far lower than the standard (Figure 2B). Meanwhile, the means of CD34, CD45, CD19, CD14, and HLA-DR negative markers are much lower than 0.5% (Figure 2B), demonstrating that the hBMSCs derived from OP patients hold intact and complete stem cells’ properties and immunomodulatory functions [28], in compliance with the international standards defined by ISCT.

The cytokines released from MSCs play a vital role in modulating the microenvironment in the MSCs-engaged tissue regeneration. In order to evaluate the capability of OP-derived hBMSCs to assist tissue regeneration, the MSCs-related critical cytokines were examined. Data reveals that VEGF and IL-6 were secreted from the OP-derived P3 hBMSCs and reached approximately 180.6 pg/mL and 122.8 pg/mL, respectively, which modulate the angiogenesis and inflammatory responses, migration, and differentiation and represent the critical cytokines secreted by a functional MSCs [29]. However, it was almost non-detected in the PDGF-BB, IL-1β, TGF-β1, IGF-1, and TNF-α secretion (Table 1). Our data demonstrate that hBMSCs derived from patients with osteoporosis would not alter cytokine secretion, which may indicate that the hBMSCs still possess intact physiological modulation capability.

### 2.4. Proliferation Capability of hBMSCs Derived from OP Patients and Juveniles and Optimization of hMSCs Culture with Serum-Free Medium

Since the cytokine secretion and surface marker representation of hBMSCs are not influenced by osteoporosis, we would like to examine further the proliferation capability of hBMSCs, which is one of the critical properties of healthy MSCs. The doubling level (DL) reflects the proliferation capability in a standard culture period, and the higher value represents a more exuberant proliferation. The OP-derived hBMSCs were cultured from P3 to P7 and compared with juvenile hBMSCs from a healthy younger donor. Data reveal that the DL was significantly diminished in the P4 and P7 osteoporotic hBMSCs compared to the juvenile hBMSCs (*p* < 0.05) (Figure 3A). However, the doubling time (DT) only significantly increased in the P7 osteoporotic hBMSCs compared with juvenile hBMSCs, which may indicate that the proliferation rate of OP-derived hBMSCs was only reduced in the higher hBMSC passage. Notably, osteoporosis does not influence P3 to P4 hBMSCs, which are conventionally used in cell transplantation in regenerative medicine. However, the hBMSCs cultured in the conventional 10% FBS/DMEM gradually decreased DL with increased DT in the serial subcultures, which may indicate a phenomenon of cell aging. In order to balance the culture expense and stimulation from the FBS-contained cytokines, we attempted to culture the hBMSCs in the P1 stage with a serum-free medium containing optimal ECM supplements and subsequently changed to the conventional FBS-contained medium in the following generation. Data reveals that the hBMSCs cultured in the serum-free medium showed a significantly higher DL than hBMSCs cultured in the FBS-contained medium in the P3 stage. However, there was almost no difference in the P5 stage. Therefore, in the following study, we standardized the hBMSC culture protocol with the serum-free medium containing optimal ECM supplements in the P1 stage, followed by a 10% FBS-contained medium, and the P3 hBMSCs were used for subsequent osteogenic and adipogenic comparison.

### 2.5. Mesodermal Differentiation Capability of hBMSCs Derived from OP Patients and Juveniles

In order to comprehensively evaluate the differentiation capability of OP patient-derived hBMSCs, the P3 hBMSCs were tested for osteogenic and adipogenic differentiation capabilities. Meanwhile, the osteogenesis and adipogenesis of OP hBMSCs were examined in parallel with the juvenile hBMSCs due to many publications demonstrating that the aging BMSCs are more prone to differentiate into adipocytes [21]. In the gross observation of osteogenesis, the OP hBMSCs and juvenile hBMSCs showed a fibroblast-like appearance at day 0, followed by flat, multi-antenna shape at days 3–4, and started to deposit white calcium zones at day 14, demonstrating the osteoblast formation (Appendix A). Similarly, both BMSCs showed identical fibroblast-like cell shape at day 0, followed by a flat, round, and hypertrophic appearance at days 4–7, and started to accumulate the white oil droplets from day 7 and became more apparent at day 20 (Appendix A).

Subsequently, qRT-PCR and calcium quantification were used to evaluate the osteogenesis of OP hBMSCs cultured with induction medium for 7 and 21 days, respectively, and compared with the juvenile hBMSCs. Our data reveal non-obvious osteogenesis in examining the Runx2 and BGLAP mRNA expression by qRT-PCR, with no significant difference between OP hBMSCs cultured with (OP-I) or without (OP-C) induction medium (*p* > 0.05) (Figure 4A,B). Nevertheless, the juvenile hBMSCs cultured with induction medium (JU-I) or without induction medium (JU-C) also showed non-significant osteogenesis (*p* > 0.05) in examining the Runx2 and BGLAP mRNA expression (Figure 4A,B), which indicates some intrinsic factors influence the osteogenic gene expression. However, the calcium examined by Alizarin Red staining at 21 days post-induction medium administration showed apparent calcium deposition in both juvenile hBMSCs and OP hBMSCs (Figure 4C). Furthermore, the quantification of deposited calcium shows a significant difference (*p* < 0.001) at post-induction 21 days at both juvenile hBMSCs and OP hBMSCs (Figure 4D), demonstrating the reserved osteogenic differentiation capability of OP patient-derived hBMSCs.

Although the qRT-PCR examination of osteogenic gene expression showed non-significant improvement after the osteogenic induction medium administration in both juvenile and OP-derived hBMSCs, we continuously examined the adipogenic differentiation capability rather than chondrogenic differentiation due to the capability of healing osteoporotic scenarios that need to be proved. Data reveal that the critical adipogenic markers FABP4 and PPARγ mRNA expression significantly increased in the juvenile and OP-derived hBMSCs after the adipogenic induction medium administration in comparison with no induction medium administration (*p* < 0.001) (Figure 5A,B). Furthermore, the deposited oil droplets of hBMSCs incubated with an induction medium were stained by Oil Red O and showed apparent oil formation at day 21 (Figure 5C). Furthermore, the quantity of oil droplet formation was examined by O.D. 492 nm absorbance, which shows significantly more abundant oil droplet formation at hBMSCs cultured with induction medium than that cultured in non-induction conditions (*p* < 0.001) (Figure 5D), demonstrating the reserved and intact adipogenic differentiation capability of OP patient-derived hBMSCs.

## 3. Discussion

Bone fractures resulting from low-energy hits and osteoporosis primarily occur in the hip joint and usually elicit life-threatening consequences. Therefore, social networks, the national medical system, and families must devote many efforts to following caring. Furthermore, a hip fracture resulting from osteoporosis is an urgent issue and a heavy burden for an aging society such as Japan and Taiwan. Meanwhile, bone natively possesses self-regeneration capability, and the hBMSCs dominate the critical role in this process. However, why can the osteoporotic patients’ hBMSCs not recover the osteoporotic pathology? Therefore, we may interpret the reason through a detailed and broad molecular mechanistic examination of the clinical hBMSCs collected from aging osteoporotic patients compared with those collected from healthy young donors. In this study, we could interpret whether osteoporotic patients’ hBMSCs lose osteoblastogenesis function, leading to bone mass regeneration malfunction and an imbalance of skeletal metabolism homeostasis. Besides, examining the fitness of hBMSCs derived from osteoporotic patients is essential because cell-based therapy using BMSCs may become one of the treatments for osteoporosis therapy in the future [10].

We collected bone marrow from 19 aging osteoporotic patients (Appendix A), established a standard hBMSC isolation, cultivation, and cryopreservation protocol, and achieved a cryopreserved hBMSC bank from 11 patients (Appendix A). Meanwhile, we conducted the proliferation test (Figure 3), osteogenic and adipogenic differentiation test (Figure 1), surface marker identification (Figure 2), and cytokine examination (Table 1) using the hBMSCs derived from the aging patients who suffered hip fractures resulting from low-energy trauma. Furthermore, the osteogenic and adipogenic differentiation and significant marker expression of hBMSCs derived from aging osteoporotic patients were quantitively compared with those from young, healthy donors (Figure 4 and Figure 5). In addition, data reveals that the hBMSCs derived from aging patients with osteoporosis highly expressed IL-6, which may play the role of immunomodulation and alter the immune circumstance of the bone cavity microenvironment (Table 1). Meanwhile, VEGF expression also increased, which may be one of the bone self-healing signals released by the microenvironment in a high bone loss scenario (Table 1). Notably, the proliferation capability of hBMSCs derived from aging osteoporotic patients compared with hBMSCs from healthy juvenile donors reveals a slight decrease in the higher passage. However, previous publications demonstrated that streptomycin used in the culture alters the proliferation and differentiation of human MSCs [30]. Our data shows that the non-induction control group is without a spontaneous differentiation scenario (Figure 4 and Figure 5). An equivalent proliferation level in passages 3–4 is conventionally used for clinical cell transplantation (Figure 3).

Nevertheless, the CD markers and osteogenic and adipogenic potentials of hBMSCs from the aging osteoporotic patients show equal capability and no significant difference compared to hBMSCs from the healthy juvenile donors (Figure 2, Figure 4 and Figure 5). Since the hBMSCs bank from the aging osteoporotic patients demonstrated multipotent differentiation capability, the interaction of hBMSCs with the microenvironment can be further explored to interpret the biological function of hBMSCs in the osteoporotic tissue. In addition, bio-macromolecular signals, such as the exosome, microRNA, and interleukin released from the OP hBMSCs, can be examined to delineate the mutual interaction between the OP hBMSCs and the bone mass microenvironment [31].

In order to prevent or reverse the gradual bone loss and increased fragility resulting from aging, scientists often emphasize how to inhibit bone resorption, such as by utilizing an anticatabolic drug. However, stimulating osteoblastogenesis and the subsequent new bone formation is more challenging to compensate for the natural aging that elicits bone deterioration [32]. A previous study echoes our current finding; Marie et al.’s interpretation is reasonable [32] because the osteogenic and anabolic dual-functioning drug has not yet been explored in detail [33,34]. However, our current data reveal that the osteogenic potential of OP hBMSCs is equivalent to the hBMSCs retrieved from healthy young donors. Nevertheless, the detailed differences have not yet been investigated comprehensively. Therefore, our hBMSCs bank collected from aging osteoporotic patients could be one of the high-throughput in vitro models for osteogenic and anabolic dual-functioned drug screening before the complex in vivo selection; it is an expensive and time-consuming experiment, even utilizing a facile zebrafish model [35]. Furthermore, previous publications using MSC high-throughput screening models successfully identified potent chondrogenic drugs [9,36].

According to previous publications, Guan et al. have developed a peptide-drug conjugate, LLP2A-Ale, which assists the transplanted BMSCs to attach to the bone tissue, simultaneously possesses osteogenic and anabolic functions to stimulate bone regeneration, and can be an alternative osteoporosis therapy [37]. Basha et al. developed a siRNA technology to silence a negative suppressor gene of BMSCs’ osteogenesis, GNAS, and successfully promote the osteoblast differentiation of BMSCs, leading to new bone formation [38]. Another study demonstrated that bisphosphonate risedronate possesses the dual function of simultaneously promoting BMSCs’ osteoblastogenesis and the consequent bony anabolic effect [33]. Kim et al. identified a new small molecule, CW008, a derivative of pyrazole–pyridine, that stimulates osteoblast differentiation of human MSCs and increases bone formation in the ovariectomized mouse model, implying the potential of developing as a new anabolic drug for osteoporosis treatment [39]. A more recent study developed a bone-targeted cerium nanoparticle drug delivery system to lead mesenchymal stem cell osteogenesis and endothelial progenitor cell angiogenesis, providing a novel anabolic therapeutic strategy for treating osteoporosis [40]. Echoing the findings and hBMSCs bank established in the present study, all novel compounds developed in the above publications can be examined and screened again in the hBMSCs derived from aging osteoporotic patients to clarify a long-termed controversial issue of whether hBMSCs withdrew from the osteoporotic patients can be applied to osteoporosis therapy? Moreover, the aging osteoporotic hBMSCs bank established in the current study can be further manufactured as a diseased organoid and developed as one of the ex-vivo cell platforms for osteogenic and anabolic drug screening for osteoporosis therapy [31,41].

On the other hand, previous findings suggested that aging or postmenopausal BMSCs hold low osteogenic potency due to a lot of BMSCs in a senescent stage need more time for propagation, favor adipogenesis, and diminish the osteoblast lifespan; meanwhile, the total number of BMSCs is also decreased [42,43]. Furthermore, the number of multipotent BMSCs and colony-forming capability is diminished in the bone marrow collected from aging populations, indicating that BMSCs’ fitness for tissue regeneration is unsatisfactory [44]. However, our study demonstrated that the multipotency and unique markers of hBMSCs collected from patients with osteoporosis aged 56 to 95 are not declined and are comparable to the hBMSCs from healthy young donors. It may be that the intrinsic endocrines essential for hBMSCs’ osteogenesis diminished, leading to the imbalanced homeostasis of a bony microenvironment and a worse osteoporosis scenario; even bone resorption remains steady. Furthermore, one critical and controversial issue is that many scientists would like to know whether the BMSCs collected from osteoporotic patients have abnormal surface receptor deployment, leading to insusceptibility to the osteogenic factors. Despite the regular expression of osteogenic factors, new bone regeneration remains inferior. To address this, we need further exploration through proteomics, single-cell RNASeq gene expression, and spatial genomic analysis to comprehensively analyze the protein and gene expression of the aging osteoporotic and young, healthy BMSCs.

Most osteoporotic medications inhibit bone resorption, including anti-osteoclast activity and anti-osteoclastogenesis drugs such as bisphosphonate and denosumab [7,8], but their therapeutic efficacy is limited. Therefore, anabolism emerged as a critical bottleneck to improving osteoporotic therapy and ameliorating bone regeneration. Although presently, few anabolic drugs have been approved by the FDA for osteoporosis therapy, such as Forteo^TM^ (teriparatide) and Evenity^TM^ (romosozumab) [45,46,47], the side effects and high price still restrict broad utilization [48]. However, in recent scientific achievements in a decade, autologous stem cells applied to tissue engineering and regenerative medicine have gained prominent advancements. As a result, the FDA has approved re-infusing them in the human body for therapy for particular diseases, such as osteoarthritis, stroke, and spinal cord injury [49]. Therefore, exploring whether autologous BMSCs can be applied to osteoporosis therapy is a practical issue. Meanwhile, if the autologous BMSC infusion is combined with osteoclastogenesis inhibition, it may open a new strategy to thoroughly change the bony microenvironment and fundamentally recover the osteoporotic pathology.

Cell therapy has become increasingly popular and acceptable for generals, and cell source and quantity have become critical factors that dominate the success rate of cell therapy. To reach a large cell quantity with consistent cell quality, higher cell passage, and consistent multipotency, it should be considered. The whole manipulation should be maintained in a GMP-grade circumstance, which is expensive and time-consuming. Researchers addressed the in vitro expansion of stem cells to reach a high quantity, revealing that senescence accompanies the passage process and potentially deteriorates stem cell fitness [50]. Besides, the tissue regeneration potential of stem cells declined with age; whether this is due to the intrinsic aging of stem cells or the impairment of stem cell function in the aged tissue micro-environment is yet fully elucidated [51]. Therefore, enriching the stem cell quantity in a higher cell passage without senescence is one of the considerations for cell therapy. Our data demonstrate the effectiveness of the hBMSCs collected from elderly patients with osteoporosis, possessing qualified multipotency, a valuable, valid, and precious stem cell source, and can serve as an alternative consideration for cell therapy.

## 4. Materials and Methods

### 4.1. Patient Recruitment Criteria in the Clinical Trials

This was a prospective, single-reader, multiple-case investigation conducted from 2019 to 2020 at one medical center, National Taiwan University Hospital at Hsin-Chu in Taiwan. Elderly patients enrolled in the current study were older than 50 years with a hip fracture resulting from low-energy trauma, termed as inclusion criteria, and were considered eligible for this study. The exclusion criteria are patients with a malignant tumor, cancer, blood disease, HIV, HBV, HCV-positive carriers, and other notifiable transmission diseases. Younger patients were selected from the healthy donors with femoral fractures resulting from transportation accidents. Patients were notified and agreed with the Informed Consent Form (ICF), and the institutional review board (IRB) committee approved the study with the approval numbers: 108-006-E and 103-018-F.

### 4.2. Isolation and Culture of Human Bone Marrow-Derived Stem Cells (hBMSCs)

Patients with hip fractures were received and primarily examined in the emergency room. Hip fracture patients who needed advanced surgery were treated in the operating room by eligible orthopedic doctors. For the extra-capsular type hip fracture, patients underwent internal fixation with their medullary canal prepared by flexible reamers. For the intra-capsular type hip fracture, patients received bipolar hemiarthroplasty with their canals enlarged by broaches. Meanwhile, the drained bone marrow was collected in the centrifuge tube containing heparin and delivered to the laboratory with bio-safety level 2 immediately for subsequent stem cell isolation. All patients who proceeded with the study recognized their consensus under the IRB protocol: 108-006-E and 103-018-F. The 20–40 mL bone marrow collected from patients was subjected to Ficoll (Cat. GE17-1440-02, Sigma-Aldrich, St. Louis, MO, USA) gradient centrifugation in 4 °C as illustrated (Appendix A). Cells in the buffy coat layer rich in BMSCs were further collected, counted, and plated in the culture dish at a density of 1 × 10^5^/cm^2^ for subsequent colony formation using α-MEM (Cat. 12571063, ThermoFisher Scientific, Waltham, MA, USA) containing 10% FBS (Cat. 12662029, ThermoFisher Scientific, Waltham, MA, USA). The medium was replaced with fresh medium every three days, and cells reached 80% confluency at approximately 10–14 days and formed foci appearance, termed P0 (Appendix A). Finally, the confluent BMSCs were trypsinized and subcultured at a density of 3.5 × 10^3^/cm^2^ for propagation. All BMSCs were cryo-preserved at the P2 stage as standard protocol for subsequent experiments in the current study.

### 4.3. Optimize the Culture Protocol of hBMSCs with Serum-Free Medium

BMSCs cultured in the complete medium α-MEM containing 10% FBS were compared with the serum-free medium (PRIME-XV^®^ MSC Expansion XSFM, Cat. 91149, FUJIFILM Irvine Scientific, Santa Ana, CA, USA) containing CELLstart^TM^ ECM (Cat. A1014201, ThermoFisher Scientific, Carlsbad, CA, USA) by examining the cell morphology, expansion, and proliferation capability. Meanwhile, the cost of both media was also considered. The P0 cells were seeded in a density of 1 × 10^5^/cm^2^ for colony formation, followed by a density of 3.5 × 10^3^/cm^2^, for subsequent culture. In addition, the doubling time of BMSCs and cumulative proliferation were evaluated. In compromise on medium cost and proliferation time, the cells harvested from the buffy coat layer were seeded and cultured using the serum-free medium for the first 24 h for adhesion, followed by medium change every three days using the complete medium, termed as standard protocol for whole subsequent experiments.

### 4.4. Proliferation Test of hBMSCs

To examine the BMSC quality, the doubling time (DT), doubling level (DL), and cumulative population doubling level (CPDL) are applied as indicators to reflect cell senescence. Passage 2 BMSCs were de-frozen and cultured in a 10-cm dish at a density of 3.5 × 10^3^/cm^2^, the medium was changed every three days, cells were harvested and counted by microchips (ADAM^TM^-MC, automated cell counter, NanoEntek, Inc., Seoul, Republic of Korea), and subcultured at day 5 and continued to passage 10. The following equation examines the proliferation capability:$$DL={log}_{10}(\frac{\mathrm{harvested cell number}\mathrm{cell}\mathrm{number}}{\mathrm{seeding}\mathrm{cell}\mathrm{number}})/{log}_{10}(2)$$
$$CPDL={DL}_{passage\;3}+{DL}_{passgge\;4}+\ldots +{DL}_{passage\;9}+{DL}_{passage\;10}$$
$$DT=\frac{\mathrm{culture}\mathrm{duration}\left(h\right)}{DL}$$

### 4.5. Quantification of Surface Markers of hBMSCs

BMSCs were collected by centrifuge and resuspended in the MACS buffer (Cat. 130-091-222, Miltenyi Biotec., Bergisch Gladbach, Germany) at a density of 6.3–8.9 × 10^5^/mL in compliance with the manufacturer’s instruction. Next, 100 μL cells were withdrawn and incubated with the primary antibody at 2–8 °C in the dark for 30 min. Primary antibodies against CD34 (1:50, Cat. 555822, BD Biosciences, San Jose, CA, USA), CD73 (1:50, Cat. 550257, BD Biosciences, San Jose, CA, USA), CD90 (1:100, Cat. 555596, BD Biosciences, San Jose, CA, USA), CD45 (1:50, Cat. 555482, BD Biosciences, San Jose, CA, USA), CD105 (1:100, Cat. MCA1557F, Bio-RAD antibodies, Hercules, CA, USA), CD19 (1:50, Cat. 555412, BD Biosciences, San Jose, CA, USA), CD14 (1:50, Cat. 555397, BD Biosciences, San Jose, CA, USA), HLA-ABC (1:50, Cat. 555552, BD Biosciences, San Jose, CA, USA) and HLA-DR (1:400, Cat. 555811, BD Biosciences, San Jose, CA, USA) were selected in the current study. Details of primary antibodies and isotypes were disclosed in the Appendix A. Subsequently, cells were centrifuged and re-suspended in 300 μL DPBS, followed by FACS analysis (model: Accuri C6, BD Biosciences, San Jose, CA, USA) according to the manufacturer’s instructions.

### 4.6. Cytokines Analysis of hBMSCs

The cell medium of passage three was collected on day 5 after initial seeding for cytokine analysis and accompanied by the proliferation test before proceeding with the other subculture. The medium was centrifuged to remove the debris, and 100 μL was subjected to ELISA measurement in compliance with the manufacturers’ instructions. ELISA kits used to detect PDGF-BB (Cat. DY220, R&D Systems, Minneapolis, MN, USA), VEGF (Cat. DY293B, R&D Systems, Minneapolis, MN, USA), IL-1β (Cat. DY201, R&D Systems, Minneapolis, MN, USA), TGF-β1 (Cat. DY240, R&D Systems, Minneapolis, MN, USA), IL-6 (Cat. DY206, R&D Systems, Minneapolis, MN, USA), IGF-1 (Cat. DY291, R&D Systems, Minneapolis, MN, USA), and TNF-α (Cat. DY210, R&D Systems, Minneapolis, MN, USA) were selected in the current study.

### 4.7. Calcium Deposition and Quantification of hBMSCs’ Osteogenesis

The passage 3 BMSCs were cultured in the 12-well plates at a fixed initial density of 1 × 10^4^/cm^2^ using a standard osteogenic induction medium composed of complete medium containing 0.1 mM dexamethasone (Cat. D2915, Sigma-Aldrich, St. Louis, MO, USA), 175 μM ascorbate-2–phosphate (Cat. A8960, Sigma-Aldrich, St. Louis, MO, USA), and 10 μM β-glycerophosphate (Cat. G9422, Sigma-Aldrich, St. Louis, MO, USA). The medium was changed every 3–4 days and continued for 28 days. Cells were fixed by 4% paraformaldehyde (Cat. sc-281692, Santa Cruz Biotechnology, Dallas, TX, USA), followed by 2% Alizarin Red S (Cat. 8678, ARed-Q, ScienCell^TM^ Research Laboratories, Carlsbad, CA, USA) staining and gross picture photography. Subsequently, the cells were washed with 1000 μL ddH_2_O and completely drained. The Alizarin Red S-stained calcium was extracted by 1 mL 10% acetic acid (Cat. W200603, Sigma-Aldrich, St. Louis, MO, USA), followed by absorbance measurement at O.D. 405 nm through a spectrophotometer (BioTek Synergy H1, Agilent, Santa Clara, CA, USA).

### 4.8. Oil Droplet Quantification of hBMSCs’ Adipogenesis

Adipogenic differentiation is almost identical to osteogenic induction with slight modification; the cell density was adjusted and fixed at 3 × 10^4^/cm^2^ at day 0, cultured with standard adipogenic induction medium containing 1 μM dexamethasone, 200 μM Indomethacin (Cat. I7378, Sigma-Aldrich, St. Louis, MO, USA), 10 μM recombinant insulin (Cat. 91077C, Sigma-Aldrich, St. Louis, MO, USA), and 0.5 mM 3-Isobutyl-1-methylxanthine (Cat. I7018, Sigma-Aldrich, St. Louis, MO, USA). The medium was changed every 3–4 days, continued to 21 days, fixed, and stained with 0.6% Oil Red O (Cat. O0625, Sigma-Aldrich, Saint Louis, MO, USA) in compliance with the manufacturer’s instruction. The Oil Red O-stained oil droplets were further extracted by 100% isopropanol (Cat. 190764, Sigma-Aldrich, St. Louis, MO, USA), followed by absorbance measurement at O.D. 492 nm through a spectrophotometer (BioTek Synergy H1, Agilent, Santa Clara, CA, USA).

### 4.9. qRT-PCR

The total RNA was extracted from the BMSCs cultured with adipogenic or osteogenic induction medium at the indicated time points revealed in the Results by the RNeasy kit (Cat. 74106, Qiagen, Germantown, MD, USA) in compliance with the manufacturer’s instructions. A 2 μg total RNA was converted to cDNA through a high-capacity cDNA reverse transcription kit (Cat. 4368814, ThermoFisher Scientific, Waltham, MA, USA), followed by the TaqMan Gene Expression Assay Mix (Cat. 4352042, ThermoFisher Scientific, Waltham, MA, USA), and real-time PCR measurements (StepOne Plus real-time PCR, Applied Biosystems™, Foster City, CA, USA). The RUNX2 (Hs01086177_m1, FAM, ThermoFisher Scientific, Waltham, MA, USA) and BGLAP (Hs01587814_g1, FAM, ThermoFisher Scientific, Waltham, MA, USA) probes were used for osteogenic measurements, and FABP4 (Hs01086177_m1, FAM, ThermoFisher Scientific, Waltham, MA, USA) and PPARγ (Hs01115513_m1, FAM, ThermoFisher Scientific, Waltham, MA, USA) were used for adipogenic measurements. The internal control is GAPDH (Hs99999905_m1, FAM, ThermoFisher Scientific, Waltham, MA, USA). The sequences of probes were retrieved from the following website (https://www.thermofisher.com/taqman/gene-expression/assay/query?keyword=&productType=ge&productSubtype=ge (accessed on 20 December 2020)).

### 4.10. Statistical Analysis

All box-dot plots are calculated from the independent experiments with N samples from biological individuals or *n* samples from independent cryopreserved vials. Moreover, it is presented as mean ± standard deviations (SDs) or mean ± standard error medium (SEM), as indicated in the results and figure captions. The statistical comparisons were performed using Student’s *t*-test or one-way ANOVA. *p* values < 0.05 were considered statistically significant and labeled as *, whereas *p* values < 0.01 and <0.001 were labeled as ** and ***, respectively. All calculations were performed using the Statistical Analysis System (SAS) licensed to China Medical University.

## 5. Conclusions

Our current data demonstrate that osteoporotic hBMSCs hold persistent osteogenic differentiation capability and are not influenced by donor aging and osteoporosis scenarios. Although the bone regeneration capability of hBMSCs is practicable, the crucial reason for osteoporosis is the degradation of bone mass, like a faucet losing its tap, which cannot stop the draining of osteoid. Therefore, further exploration through spatial proteomic and genomic analysis to delineate the molecular scenario and signal transduction mechanism of hBMSCs in the osteoporotic bony microenvironment is necessary for a breakthrough in osteoporosis therapy.

## Figures and Tables

**Figure 1 ijms-25-05273-f001:**
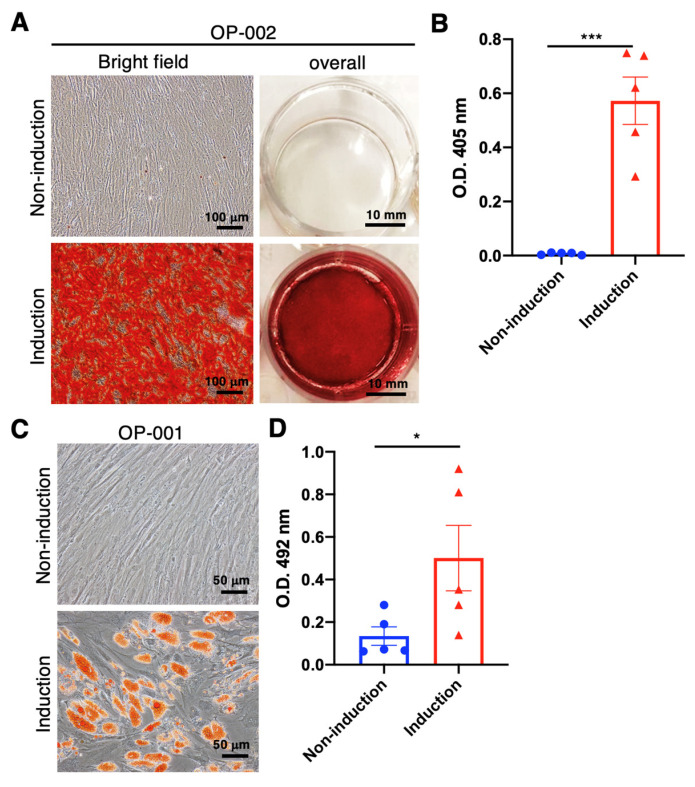
Osteogenic and adipogenic induction using the BMSCs collected from aging and osteoporotic patients with low-energy traumatic hip fractures for self-comparison of osteogenic and adipogenic differentiation capability. (**A**) In representative osteogenic images, BMSCs were stained with Alizarin-Red at day 28 after administering the osteogenic medium. (**B**) Quantitively examine the Alizarin-Red-stained calcium deposition. (**C**) Representative adipogenic images and BMSCs were stained with Oil-Red at day 21 after adipogenic medium administration. (**D**) Quantitively examine the Oil-Red-stained oil drop formation. (N = 5, data points represent biological individuals OP-001, OP-002, OP-003, OP-009 and OP-010), * *p* < 0.05, *** *p* < 0.001, data represented as mean ± SD.

**Figure 2 ijms-25-05273-f002:**
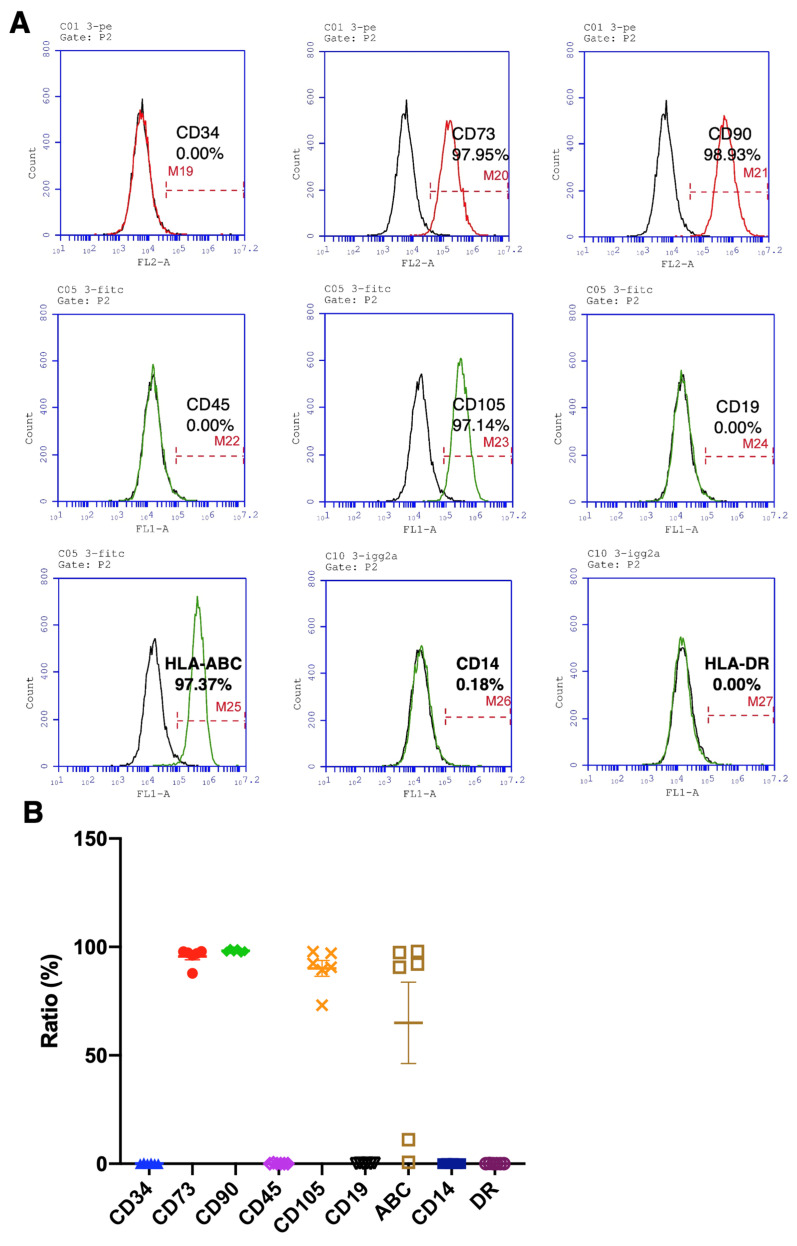
Quantitively examine the specific markers of osteoporotic BMSCs by flow cytometry for stem cell characteristics evaluation. (**A**) Representative histograms from OP-016 show the percentages of indicated cellular markers. (**B**) Percentage representation of all BMSC markers examined in the current study (N = 6 biological individuals OP-003, OP-004, OP-009, OP-014, OP-016, and OP-017). Data represented as mean ± SEM.

**Figure 3 ijms-25-05273-f003:**
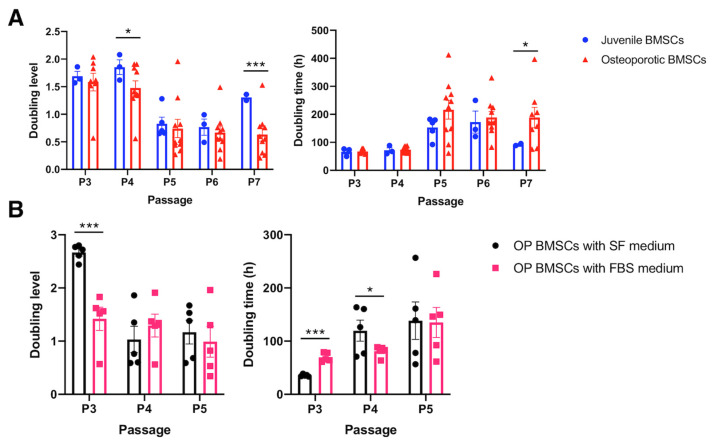
Cell proliferation assay to access the BMSCs’ proliferation capability. (**A**) Quantitively examine the cell doubling level and doubling time of the osteoporotic BMSCs compared to the young healthy juvenile BMSCs. (Juvenile N = 3 biological individuals; OP N = 8 biological individuals). (**B**) Quantitively examine the cell doubling level and doubling time of the osteoporotic BMSCs cultured by traditional medium containing 10% FBS and optimized serum-free medium. (N = 5 biological individuals). * *p* < 0.05, *** *p* < 0.001, data represented as mean ± SEM.

**Figure 4 ijms-25-05273-f004:**
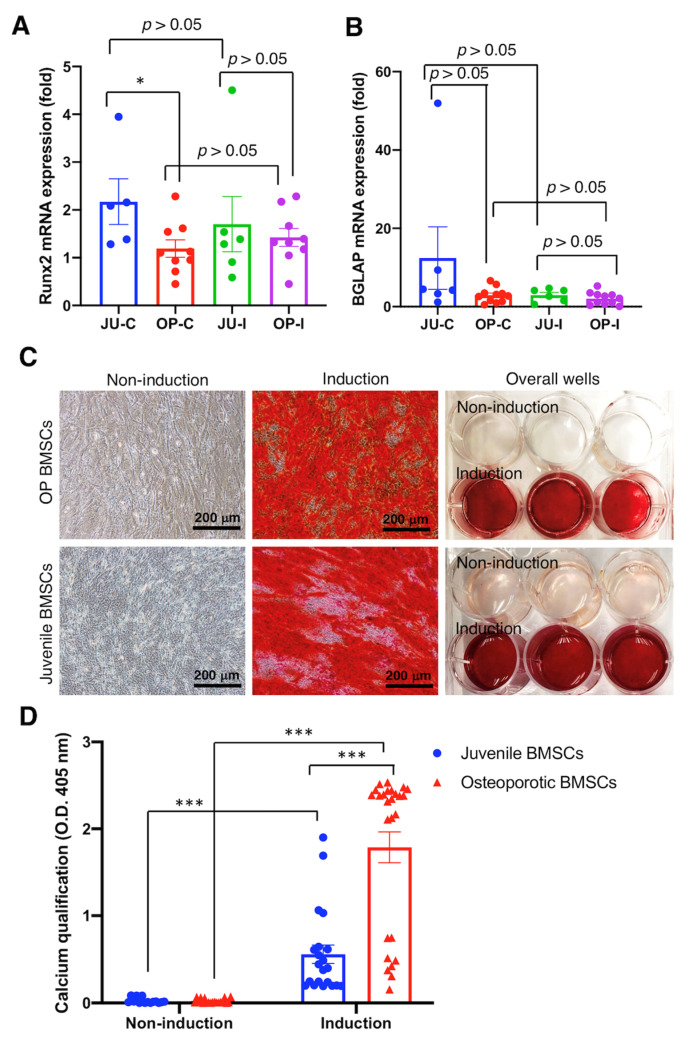
Osteogenic differentiation of osteoporotic BMSCs compared with BMSCs from young, healthy donors to access the intact osteogenic differentiation capability of OP BMSCs. (**A**) Runx2 osteogenic gene expression was analyzed by qRT-PCR. (**B**) BGLAP osteogenic gene expression was analyzed by qRT-PCR. ((**A**,**B**) Juvenile N = 3 biological individuals; OP N = 8 biological individuals) (**C**) Alizarin-Red staining. (**D**) Quantitively examine the Alizarin-Red deposition. ((**C**,**D**) *n* ≥ 10 technical replicates from OP-003 and T2B-007 individuals in independent experiments) * *p* < 0.05, *** *p* < 0.001, data represented as mean ± SEM.

**Figure 5 ijms-25-05273-f005:**
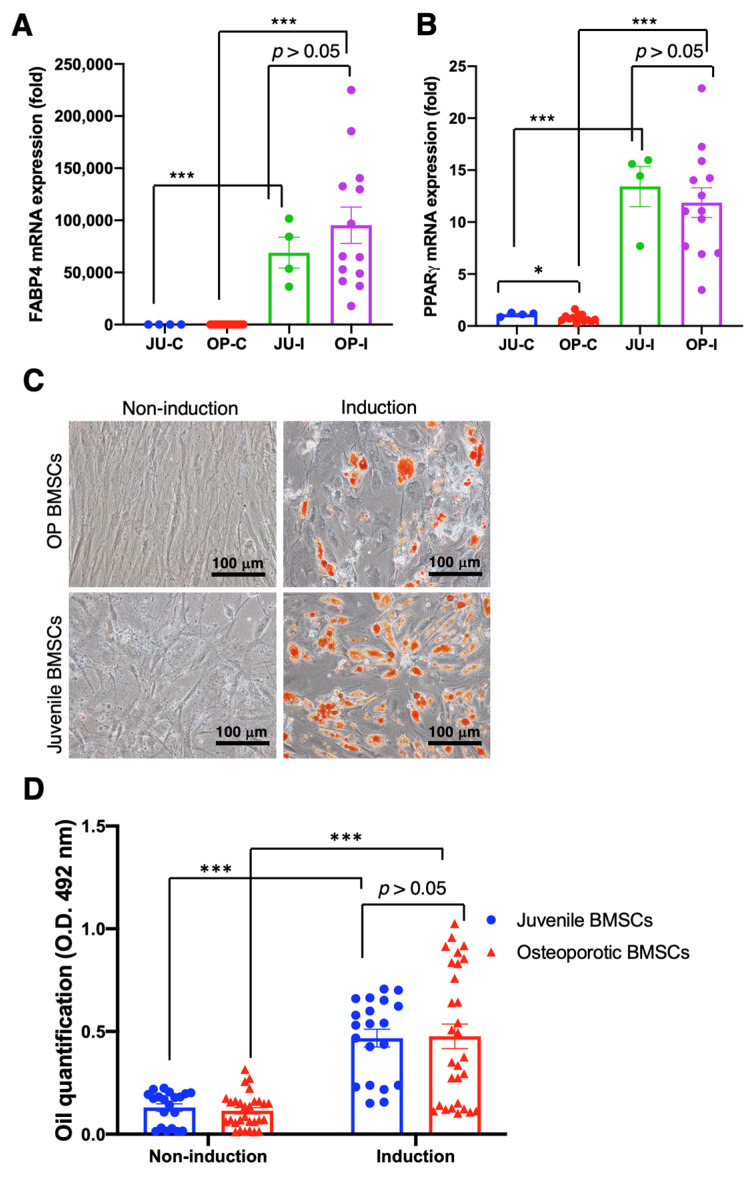
Adipogenic differentiation of osteoporotic BMSCs compared with BMSCs from young, healthy donors to access the intact adipogenic differentiation capability of OP BMSCs. (**A**) FABP4 adipogenic gene expression was analyzed by qRT-PCR. (**B**) PPARγ adipogenic gene expression analyzed by qRT-PCR. ((**A**,**B**) Juvenile N = 3 biological individuals; OP N = 8 biological individuals) (**C**) Oil-red staining. (**D**) Quantitively examine the Oil-Red deposition. ((**C**,**D**) *n* ≥ 10 technical replicates from OP-003 and T2B-007 individuals in independent experiments) * *p* < 0.05, *** *p* < 0.001, data represented as mean ± SEM.

**Table 1 ijms-25-05273-t001:** Cytokines expressed from OP patients’ BMSCs.

Cytokine (pg/mL)\BMSC	OP-001	OP-002	OP-003	OP-004	OP-009	OP-010
PDGF-BB	-	-	-	-	-	-
VEGF	172.2	172.8	125.6	190.2	132.7	290.2
IL-1β	-	-	-	-	-	-
TGF-β1	-	-	-	-	-	-
IL-6	50.36	78.83	128.77	371.23	21.73	86.14
IGF-1	s	-	-	-	-	-
TNF-α	-	-	-	-	-	-

-: undetectable, below the sensitivity of ELISA measurement.

## Data Availability

Data are shown in the Results and Appendix A and are available upon request.

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
