# Peer review of "Persistent Mesodermal Differentiation Capability of Bone Marrow MSCs Isolated from Aging Patients with Low-Energy Traumatic Hip Fracture and Osteoporosis: A Clinical Evidence"

_ijms, 2024, doi:10.3390/ijms25105273_

Round 1

Reviewer 1 Report

Comments and Suggestions for Authors

Interesting topic on whether bone marrow from healthy young and old osteoporotic adults present similar  CD markers, cytokines, and osteogenic and adipogenic differentiation.

Revise the abstract as it isn't always clear.

I would like for the authors to include the chondrogenic differentiation of MSC to have a full comparison between young and healthy patients and old and osteoporotic adults.  

Line 486, check the units of measure.

Line 457, how did you count the cells?

Did you use isotype controls for flow cytometry?

Line 489 did you wash the cells after staining and before Alizarin S extraction?

All the information of reagents must be detailed. Anyone should be able to purchase the exact same product you have used. 

Was the medium above the cells taken at day 5 after initial seeding? 

Comments on the Quality of English Language

Minor English mistakes.

Author Response

We appreciate the valuable comments from the respected reviewer. The responses are attached as follows.

Reviewer 2 Report

Comments and Suggestions for Authors

Elderly patients who suffer low-energy traumatic bone fractures are characterised by bone fragility and catabolism, and ongoing osteoporosis. This manuscript sought to determine whether the BMSCs derived from the patients who suffered low-energy traumatic hip fracture preserved mesodermal differentiation capacity into the osteogenic lineage. The manuscript is generally well written and the results relevant. Moreover, cell passage number was controlled for, which enhanced scientific rigour. However, questions arise with regards to the method of data compilation for the figures.

Major

Generally, the figures do not provide sufficient explanation to give a complete understanding of what is being shown or compared. This needs to be improved before publication.

The information provided in the methods is also insufficient to give a better picture: “all box-dot plots are calculated from the independently repeated experiments (N values) and repeated specimens (values)” that could mean any of several manners of tallying the data. Does repeated “experiments” and “specimens” refer to technical and biological replicates? Given the number of samples (11 osteoporotic and 3 young) in the context of the data points apparent in the figures there appears to have been a mixing of the two. To avoid confusion, the exact number of both biological and technical replicates should be given in each figure legend.

Figure 1

Figure 1 is redundant to Figures 4 and 5 panels D. Is this repeated data? If so, it should be consolidated into the later figures. Why does the OD for osteogenic detection change from 450 to 405 for Figure 1B and Figure 4D?

Panels A and C pertain to two osteoporotic hBMSC samples (OP-001 & OP-002). Is representability shown in panels B and D? Does N=5 refer to biological or technical replicates? Provide in the figure legend an explanation of what data points in panels B and D represent.

For instance: Each data point represents the mean of X technical replicated from Y biological replicates. Or, something to the effect.

The figure would garner greater relevance if compared to the younger hBMSC samples.

Table 1

Table 1 is incorrectly annotated. There is no * to provide context to the note: “* Note: - means undetectable below the sensitivity of ELISA measurement.” An “s” also appears in the table which is not explained in the legend.

Figure 2

Figure 2 requires greater elaboration. In panel A are the histograms generated from multiple donors or an individual donor? If multiple, which donors? Does panel B show the results from 6 of the osteoporotic hBMSC samples. What were the results of the other five osteoporotic hBMSC samples?

Explain the significance of HLA-ABC being present on the surface of osteoporotic hBMSC samples. Provide a reference.

Figure 3

Figure 3: In panels A and B there are between 8-10 data points per passage condition for the osteoporotichBMSC samples (red histogram bars). Do these represent individual samples? If so, what happened to the missing samples?

Offer an explanation as to how for passage 4 (P4) doubling level can achieve statistical significance, whereas doubling time does not?

At times there are five juvenile BMSCs data points per passage condition, when there were only three samples originally collected. How is this? Were some samples duplicated? If so, why and how was the determination made of which to repeat?

Figure 4

Figure 4: In panels A & B there are between 5 and 6 juvenile data points per condition. Do some represent duplicates of the same sample? How many independent donor samples were employed in this analysis. The same question applies to the OP hBMSCs samples.

Figure 5

Figure 5: Again, indicate what the data points represent in biological and technical replicates.

Minor

Figure legends contain grammar errors that need to be addressed.

This sentence requires adjustment as it seems contradictory in logic: “Our data demonstrate that patients with osteoporosis would not alter the cytokine secretion of hBMSCs, which may indicate that the hBMSCs still possess differentiation capability.

It is not mentioned whether antibiotics were used in the growth and expansion of the cells. This should be included for the sake of reproducing the results. If streptomycin was employed, then some discussion should be included concerning the emerging evidence that the presence of streptomycin may alter cell proliferation and differentiation along certain cell lineages.

Comments on the Quality of English Language

The quality of the written English is generally good. Minor typographical and grammar errors exists that should be addressed. 

Author Response

(The authors gave the same response as above.)

Round 2

Reviewer 1 Report

Comments and Suggestions for Authors

The authors didn't fully followed my instructions.

The authors must improve the following:

All the reagents must indicate the manufacturer, city and Country, e.g. FBS, salts. Please, carefully read the manuscript and provide the information.

There are syntax and grammar mistakes in the text. "Mediums"?

Fir the used Ab, specify the used dilutions.

Comments on the Quality of English Language

Check the syntax and grammar
